**Dynamic path dependent landslide susceptibility modelling**
Jalal Samia [1, 2, 3], Arnaud Temme [4, 5], Arnold Bregt [1], Jakob Wallinga [2], Fausto Guzzetti [6], Francesca Ardizzone [6]
[1] Laboratory of Geo-Information Science and Remote Sensing, Wageningen University & Research, 6708 PB
Wageningen, Droevendaalsesteeg 3, The Netherlands
[2] Soil Geography and Landscape group, Wageningen University & Research, 6708 PB, Wageningen,
Droevendaalsesteeg 3, The Netherlands
[3] Department of Geography and Urban Planning, University of Mazandaran, 47416-13534 PB, Babolsar, Pardis
Campus, Iran
[4] Department of Geography, Kansas State University, 920 N17th Street, Manhattan, KS, 66506, United States
[5] Institute of Arctic and Alpine Research, University of Colorado, Campus Box 450, Boulder, CO 803309-0450,
Colorado, United States
[6] Istituto di Ricerca per la Protezione Idrogeologica, Consiglio Nazionale delle Ricerche, via Madonna Alta 126,
06128 Perugia, Italy
**Correspondence**: Jalal Samia, jalal.samia@wur.nl, +31617699436, +31 (0)317 – 419000
**Abstract**
This contribution tests the added value of including landslide path dependency in statistically-based landslide
susceptibility modelling. A conventional pixel-based landslide susceptibility model was compared with a model
that includes landslide path dependency, and with a purely path dependent landslide susceptibility model. To
quantify path dependency among landslides, we used a Space-Time Clustering (STC) measure derived from
Ripley's space-time K function implemented on a point-based multi-temporal landslide inventory from the
Collazzone study area in central Italy. We found that the values of STC obey an exponential decay curve with
characteristic time scale of 17 years, and characteristic space scale of 60 meters. This exponential space-time decay
of the effect of a previous landslide on landslide susceptibility was used as the landslide path dependency
component of susceptibility models. We found that the performance of the conventional landslide susceptibility
model improved considerably when adding the effect of landslide path dependency. In fact, even the purely path
dependent landslide susceptibility model turned out to perform better than the conventional landslide susceptibility
model. The conventional plus path dependent and path dependent landslide susceptibility model and their resulted
maps are dynamic and change over time unlike conventional landslide susceptibility maps.

**1. Introduction**
Landslide susceptibility modelling calculates the likelihood of landslide occurrence in a certain location (Brabb,
1985). The resulting landslide susceptibility maps from landslide susceptibility models indicate where landslides
are likely to occur (Guzzetti et al., 2005). These maps are useful in land use planning and insurance, among others.
In this context, different methods and techniques have been used for landslide susceptibility modelling.
Reichenbach et al. (2018) classified these methods and techniques into five groups: (i) direct geomorphological
mapping, (ii) analysis of landslide inventories, (iii) heuristic or index-based approaches, (iv) physically or process-
based methods, and (v) statistically-based techniques.
Statistically-based landslide susceptibility techniques have been the preferred technique in the modelling of
landslide susceptibility (Reichenbach et al., 2018). In statistical landslide susceptibility modelling, empirical
quantitative relations are explored between the spatial distribution of landslides and a set of environmental factors
(e.g., slope and geology) (Van Westen et al., 2003; Guzzetti et al., 2005). The spatial distribution of historic
landslides, documented in landslides inventories, is therefore a crucial input for statistically-based landslide
susceptibility modelling (Guzzetti et al., 2012; Van Westen et al., 2008). Direct field mapping, visual interpretation
of aerial photographs and other remote sensing images are the main sources for such mapping of landslide
inventories (Guzzetti et al., 2012). Landslides in such inventories are stored as points or polygons. Although
polygon-based landslide inventories (Ardizzone et al., 2018; Schlögel et al., 2011; Galli et al., 2008) are becoming
increasingly available, in many landslide prone regions only less-detailed point-based landslide inventories are
collected (Gorum et al., 2011; Sato et al., 2007; Keefer, 2000). Conditioning attributes used in landslide
susceptibility modelling are mainly derivatives of digital elevation models (DEMs) along with geological, soil and
land use data (Günther et al., 2014; Neuhäuser et al., 2012; Reichenbach et al., 2018). While geology, land use and
soil data are not always available in high detail, DEM-derivatives are easily computed and globally available at a
range of resolutions. Therefore, the minimum available dataset for landslide susceptibility modelling includes a
point-based landslide inventory and a set of DEM-derived conditioning attributes.
Traditionally, landslide susceptibility is considered time-invariant: susceptibility of a place to landslide occurrence
is constant over time, at least over decadal scales. Recently, we proposed the concept of time-variant landslide
susceptibility, where susceptibility changes over time due to the transient effect of previous landslides on future
landslide occurrence (Samia et al., 2017b, a). We referred to such a transient effect as "path dependency", a term
adopted from complex system theory where it is used to describe the concept that the history of a system specifies
the future behaviour of a system through legacy effects (Phillips, 2006). In our study area in Umbria, central Italy
(Figure 1), we identified the existence of path dependency among landslides: earlier landslides locally increased
the susceptibility for future landslides for about two decades, during which the susceptibility decays exponentially
over time (Samia et al., 2017b). We first implemented the effect of this landslide path dependency in landslide
susceptibility modelling at the scale of slope units. Such units divide an area into hydrological units bounded by
drainage and divide lines (Carrara et al., 1991; Alvioli et al., 2016). We found that the impact of path dependency
on landslide susceptibility models at slope-unit scale was limited (Samia et al., 2018). This limited impact of
landslide path dependency on model predictions was probably due to the fact that landslide path dependency
affects landslide patterns at spatial scales smaller than slope units, and we hypothesized that differences between
models were likely to increase when including path dependency at smaller spatial scales.
The objective of this work is thus to consider the effect of landslide path dependency in landslide susceptibility
modelling at the resolution of $10 \times 10$ m pixels. We hypothesize that including landslide path dependency will
improve the performance of landslide susceptibility models. We also explore whether a purely path dependent
landslide susceptibility model, i.e. based solely on landslide inventory information, can provide a meaningful
landslide susceptibility map. We use the unique multi-temporal landslide inventory from the Collazzone study area
(Figure 1) (Guzzetti et al., 2006a; Ardizzone et al., 2007; Ardizzone et al., 2013).

## 2. Study area and data

The Collazzone study area, Umbria, central Italy (Figure 1), extends for about 80 km$^2$ with terrain elevation
between 140 to 632 m and terrain slope derived from a $10 \times 10$ m DEM (Figure 2) between 0 to 64°. The DEM
was prepared by interpolating 5- and 10-m contour lines shown in 1:10,000 topographic maps (Guzzetti et al.,
2006b). A set of DEM-derivatives that has been widely used in landslide susceptibility modelling was computed
in SAGA GIS and ArcGIS. We expect that these DEM-derivatives capture topographical, geomorphological and
hydrological properties of locations in our study area.
The DEM-derivatives (Figure 2) are slope angle, curvature, plan and profile curvature, aspect, northness and
eastness as cosine and sine transformations of aspect, topographic position index (TPI) representing different
geomorphological settings (Costanzo et al., 2012), stream power index (SPI) representing the erosive power of
streams (Moore et al., 1993), topographic wetness index (TWI) as an index for hydrological process in the slope
(Jebur et al., 2014). Additionally, for every pixel we computed the distance to the nearest river, the slope length
and steepness factor (LS factor) as an index for soil erosion on slope (Moore and Wilson, 1992), the vertical
distance to the slope's channel network, and relative slope position representing the relative position of slope in
cells between the valley bottom and ridgetop. Additionally, we calculated topographic roughness, which expresses
the difference in the values of elevation in the neighbouring cells in the DEM (Riley et al., 1999), and the standard
deviation of elevation and slope in a $3 \times 3$ pixel window. These 16 DEM-derivatives were used as independent
explanatory variables in logistic regression for modelling of landslide susceptibility (see section 3.2).
Landslides are abundant in this area, and range from recent shallow landslides to old deep-seated landslides
(Guzzetti et al., 2006a). Intense and prolonged rainfall and rapid snowmelt are the main triggers of landslides in
the area (Cardinali et al., 2000; Ardizzone et al., 2007). A unique multi-temporal landslide inventory with 3391
landslides has been mapped in 19 different time slices. The age of the landslides ranges from relict and very old
landslides with an uncertain date of occurrence to landslides that have occurred in 2014. Aerial photographs, direct
geomorphological field mapping and satellite images were used for the preparation of the multi-temporal landslide
inventory (Ardizzone et al., 2013; Guzzetti et al., 2006a; Galli et al., 2008). Only time slices of the multi-temporal
inventory for which the relative date of occurrence is known (Figure 1), were used in this study because time
between landslides is a key element in the quantification of landslide path dependency (Samia et al., 2017a, b). In
addition, the first time slice, with the known date of 1939, was only used in the computation of the landslide path
dependency parameters, and not in landslide susceptibility modelling because of its unknown past. Ultimately, a
multi-temporal landslide inventory was used that contains distribution of landslides in16 time slices dating from
1947 to 2014 (Figure 1). This multi-temporal landslide inventory was mostly prepared at the scale of 1:10,000
which is sufficient for conversion to a $10 \times 10$ m pixel-based landslide inventory. However, time slices from 1939
to 1997 were prepared from aerial photographs with scales ranging from 1:15,000 to 1:33,000, and this may
introduce some positional inaccuracy in landslides, in the order of one pixel. Given that the median size of landslide
in this period is 19 pixels, we believe that this is an acceptable level of inaccuracy.
More information about the Collazzone study area and the multi-temporal landslide inventory is given in (Galli et
al., 2008; Guzzetti et al., 2006a; Guzzetti et al., 2009; Ardizzone et al., 2007).

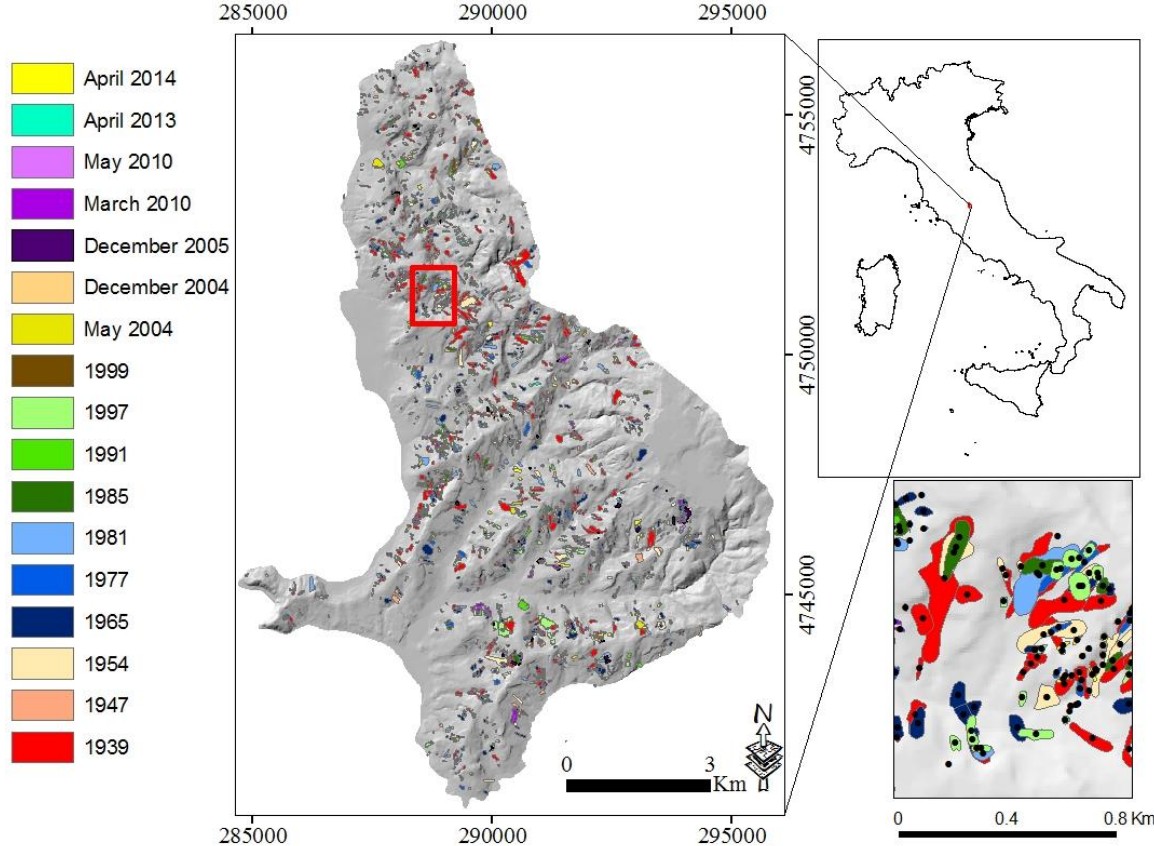


**Figure 1.** Multi-temporal landslide inventory dating from 1939 to 2014 (left map) (adapted from (Samia et al.,
2018; Samia et al., 2017a, b)). Collazzone study area and Umbria region (right upper map). The coordinate system
of maps is EPSG:32633 (www.spatialreference.org). Landslide points were constructed by placing a point in the
geometric centre of each landslide polygon (map in the right lower corner). The red rectangle shows the extent of
the map in the lower right.

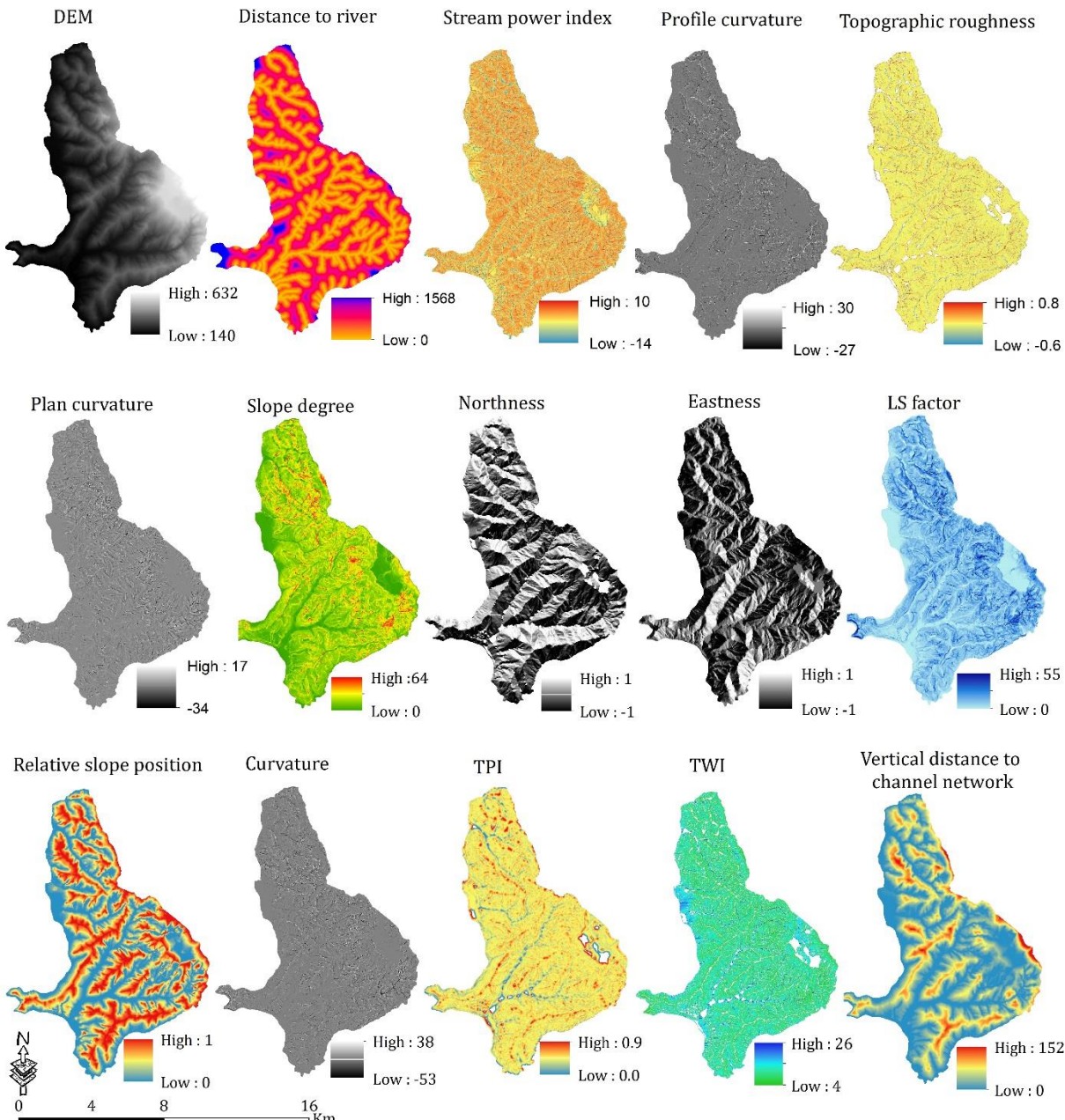

**Figure 2.** DEM (digital elevation model) and its derivatives used in conventional and conventional plus path dependent landslide susceptibility models. TPI means topographic position index, TWI means topographic wetness index and LS factor stands for slope length and steepness factor.

**3. Methods**

We used logistic regression to construct three different landslide susceptibility models (Figure 3): (i) a conventional landslide susceptibility model using DEM-derivatives, (ii) a conventional plus path dependent landslide susceptibility model using 16 DEM-derivatives and two landslide path dependency variables (explained below), and (iii) a purely path dependent landslide susceptibility model using only the two landslide path dependency variables. We compared the performance of these models using Area Under Curve (AUC) values from the Receiver Operating Characteristic (ROC) (Mason and Graham, 2002), and selected the optimal model using

the Akaike Information Criterion (AIC) (Akaike, 1998), which penalizes the use of additional variables in a model.
Ultimately, the coefficients of the variables selected by three landslide susceptibility models and the resulting
landslide susceptibility maps were compared.

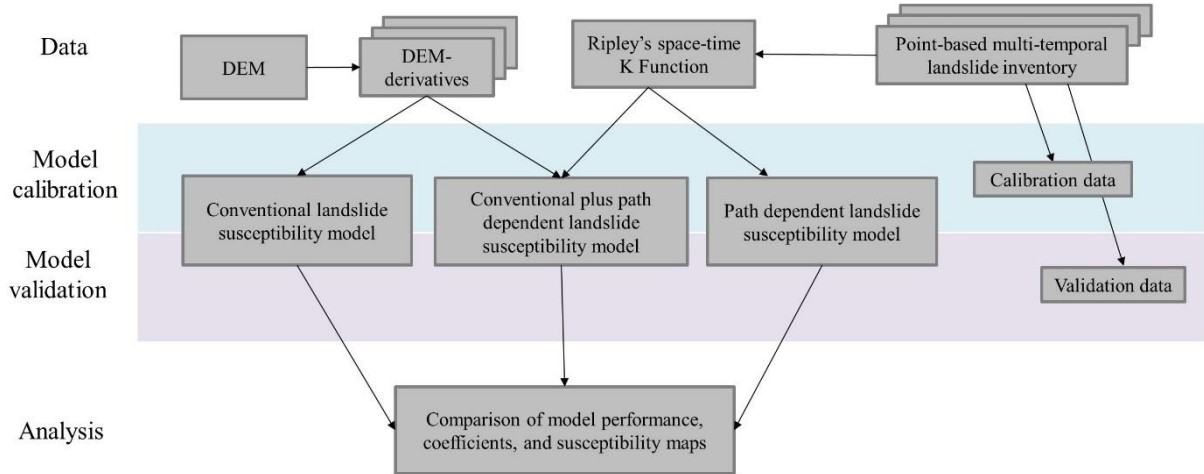

**Figure 3.** Flowchart of methods

**3.1 Quantifying landslide path dependency using Ripley's space-time K function**
The spatial-temporal dynamics of landslide path dependency was recently quantified for the Collazzone study area
(Samia et al., 2017a), and was implemented in landslide susceptibility modelling at the scale of slope units (Samia
et al., 2018). Our previous quantification of landslide path dependency used simplified information about the
spatial overlap among landslides in a polygon-based multi-temporal landslide inventory (Samia et al., 2017b). The
novel aspect of the present paper is that now, at finer spatial resolution, we quantify landslide path dependency
simultaneously in space and time. For this quantification, we use Ripley's K function (Ripley, 1976; Diggle et al.,
1995). Ripley's K function has been used mainly in spatial point pattern analysis and reflects the degree of spatial
clustering of events (e.g., landslides (Tonini et al., 2014), forest fire (Gavin et al., 2006), crimes (Levine, 2006)
and disease outbreaks (Hinman et al., 2006)). The function determines whether events are clustered, dispersed or
randomly distributed. A modified Ripley's K function was also used to quantify the degree of clustering of point
events in space and time (Lynch and Moorcroft, 2008; Ye et al., 2015). In the landslide path dependency context,
we used Ripley's space-time K function to reflect the degree to which landslides occur near previous landslides,
and how this changes with increasing distance to the previous landslide in space and time. The starting point to
derive Ripley's K is a point-based multi-temporal landslide inventory consisting of points in the geometric centre
of polygons of landslides (Figure 1).
Ripley's space-time K function tests whether the number of events that is observed in a space-time cylinder around
an initial event is equal to what is expected given the average point density in space and time (Ripley, 1976, 1977;
Diggle et al., 1995). The space-time cylinder $I_{(h, \Delta)}$ (Figure 4) is defined as:
$$I_{(h,\Delta)}\left(d_{ij}, t_{ij}\right) = \begin{cases} 1, \left(d_{ij} \leq h \text{ and } \left(t_{ij} \leq \Delta\right)\right) \\ 0, otherwise \end{cases} \qquad (1)$$

where $h$ shows the spatial distance increment, $\Delta$ shows the time increment, $i$ and $j$ are two landslide centre points, and $d$ and $t$ reflect the distance and time between the two landslide centre points, respectively.

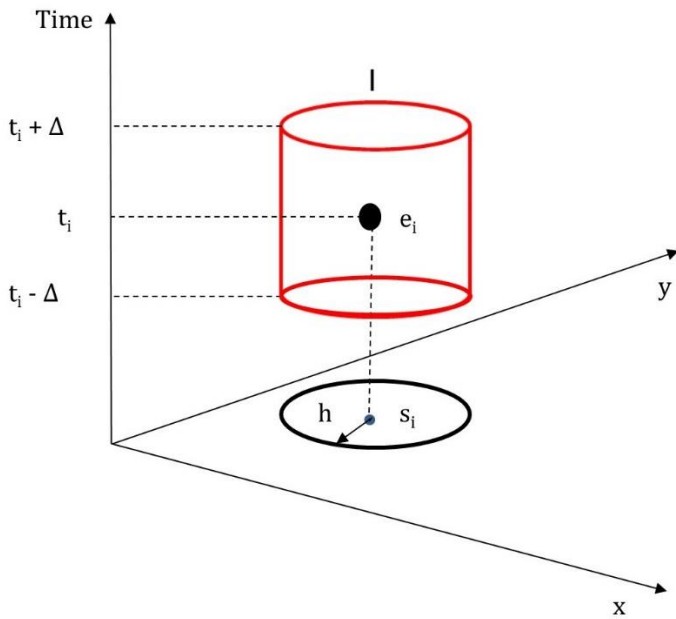

**Figure 4.** Space-time cylinder neighbourhood (Smith, 2016) for a landslide event ($e_i$)

The expected Ripley's K function for one space-time cylinder of size $h$ and $\Delta$ is defined as:

$$K(h,\Delta) = \frac{1}{\lambda_{st}} \sum_{j \neq i} E\left[I_{(h,\Delta)}\left(d_{ij}, t_{ij}\right)\right] \qquad (2)$$

where E is the expected number of landslides in the cylinder, and $\lambda_{st}$ reflects the average space-time intensity of the landslides i.e., the expected number of landslides per unit of space-time volume, which is calculated as:

$$\lambda_{st} = \frac{n}{a(R) \times (t_{max} - t_{min})} \qquad (3)$$

where $n$ is the number of landslides in the entire inventory, $t$ is time, and $a(R)$ reflects the size of the area. Therefore, the expected Ripley's space-time K function for the space-time cylinders around each landslide point is defined as:

$$K(h,\Delta) = \frac{1}{n.\lambda_{st}} \sum_{i=1}^{n} \sum_{j \neq i} E\left[I_{(h,\Delta)}\left(d_{ij}, t_{ij}\right)\right] \qquad (4)$$

Similarly, the observed Ripley's space-time K function is calculated from the landslide inventory as:

$$\widehat{K}(h,\Delta) = \frac{1}{n.\widehat{\lambda}_{st}} \sum_{i=1}^{n} \sum_{j \neq i} I_{(h,\Delta)}\left(d_{ij}, t_{ij}\right). \qquad (5)$$

Finally, we defined the space-time clustering (STC) measure, which reflects how much more likely it is that a landslide will occur given a time and space distance from a previous landslide, as following:

Empirical $STC(h,\Delta) = \frac{\widehat{K}(h,\Delta)}{K(h,\Delta)} - 1 \qquad (6)$

STC values > 0 indicate clustering and values < 0 indicate dispersion. We calculated STC (h, Δ) for a wide range
of *h* and Δ: values of h ranged from 0 to 500 meter in 30 steps, and values of Δ ranged from 0 to 38 years in 30
steps. This yielded 900 empirical values of STC (h, Δ). We then fitted an exponential decay function of *h* and Δ to
the empirical STC values. This exponential decay function was used to calculate STC values for each pixel
depending on when and where a landslide last occurred closely to that pixel.
Based on this, we calculated two landslide path dependency variables (Figure 5). The first variable reflects the
maximum value of all STC values for all previous landslides near a pixel. This variable results in high values when
one particular previous nearby landslide is expected to have a large impact on the susceptibility of landsliding. The
second variable is the sum of all STC values of all previous landslides near a pixel. This variable results in high
values when all previous nearby landslides are expected to have a large impact on the susceptibility of landsliding.
This approach mirrors what we did in our slope unit-based susceptibility model (Samia et al., 2018) in the sense
that the variables separate the impact of the most impactful previous nearby landslide from the impacts of all
previous nearby landslides.

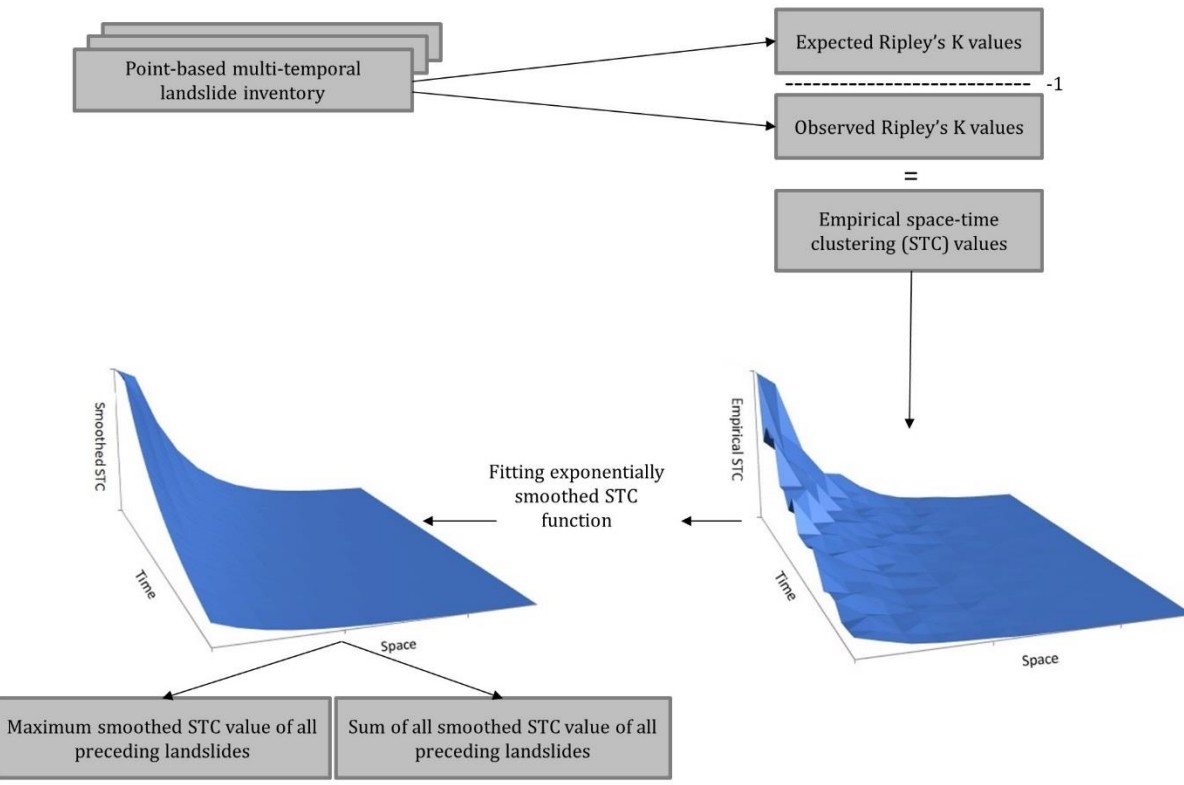

**Figure 5.** Procedure to compute the two landslide path dependency variables using Ripley's space-time K function.
**3.2 Logistic regression**
Logistic regression is considered a reference model in statistically-based landslide susceptibility modelling
(Reichenbach et al., 2018). Relations between presence and absence of landslides as a binary target variable are
explained by a set of independent variables such as slope steepness and slope position in logistic regression. In
this paper, DEM-derivatives (section 2 and Figure 2) as well as the two landslide path dependency variables
computed using the Ripley's space-time K function (see section 3.1) were used as independent variables. Landslide
presence or absence was the binary target variable.

### 3.3 Training and testing

When using a multi-temporal landslide inventory in landslide susceptibility modelling, the selection of time slices
for the training and testing is crucial. In Rossi et al. (2010) and Samia et al. (2018), a sequential splitting sampling
strategy was used in such a way that landslides in older time slices were used to train the model and landslides in
newer time slices were used to test the model. However, such a sequential sampling strategy does not provide an
equal range of landslide histories between training and testing datasets and this could bias the role of time in path
dependent landslide susceptibility modelling. To avoid such a timing inequality, Samia et al. (2018) also
introduced a non-sequential sampling strategy in which the span of timing segregation among time slices in the
training and the testing datasets is comparable. In this study, we used this sampling strategy to split the multi-
temporal landslide inventory into training and testing datasets. To achieve this, all landslides in the time slices of
1947, 1954, 1981, 1985, 1999, May 2004, March and May 2010 were used for training, and all landslides in the
time  slices of 1965, 1977, 1991, 1997, December 2004 and 2005 and April 2013 and 2014 were used for testing
(Figure 1). It is important to note that the time slice in 1939 was used only for quantification of landsliding history
of the other time slices, and not for training or testing. Thus, the $1^{st}$ time slice in the training dataset is 1947 (Figure

207  1).

The number of pixels with landslides was smaller than the number of pixels without landslides in both training
and testing datasets. Therefore, we randomly selected 5,000 pixels with landslides and 5,000 pixels without
landslides from both training and testing datasets in order to create equal datasets both for training and testing of
the models. This random selection of pixels was repeated 10 times both in the training and testing datasets.
Therefore, we trained the conventional, conventional plus path dependent and purely path dependent landslide
susceptibility 10 times, and finally tested 10 times as well. After preparation of the 10 training datasets, logistic
regression was applied to the10 training datasets with entry probability of 0.05 and removal probability of 0.06 for
independent variables to diminish the chance of overfitting in the model. We only allowed inter-variable
correlations less than 0.8 to avoid multicollinearity. Conventional landslide susceptibility was modelled using
DEM-derivatives only once for the defined training dataset and was tested using the independent testing dataset.
Conventional plus path dependent landslide susceptibility model was constructed using DEM-derivatives plus the
two landslide path dependency variables. The purely path dependent landslide susceptibility was modelled only
by using the two landslide path dependency variables. All three models were constructed only once. Model
performance was assessed using AUC and AIC values. The AUC values for testing were assessed using 10 training
models and 10 independent testing datasets. The models with highest performance in terms of AUC values, were
used to map susceptibility to landslides. Finally, we compared landslide susceptibility maps resulting from
conventional, conventional plus path dependent and purely path dependent susceptibility.


## 4. Results

### 4.1 Spatial-temporal dynamic of landslide path dependency

Ripley's space-time K function confirmed the existence of landslide path dependency at small spatial and small temporal distances from a previous landslide (Figure 6). The STC measure (Eq 6) is high in the space-time vicinity of an earlier landslide, and it then decreases rapidly. Apparently, landslide susceptibility is relatively high immediately after occurrence of an earlier, nearby landslide.

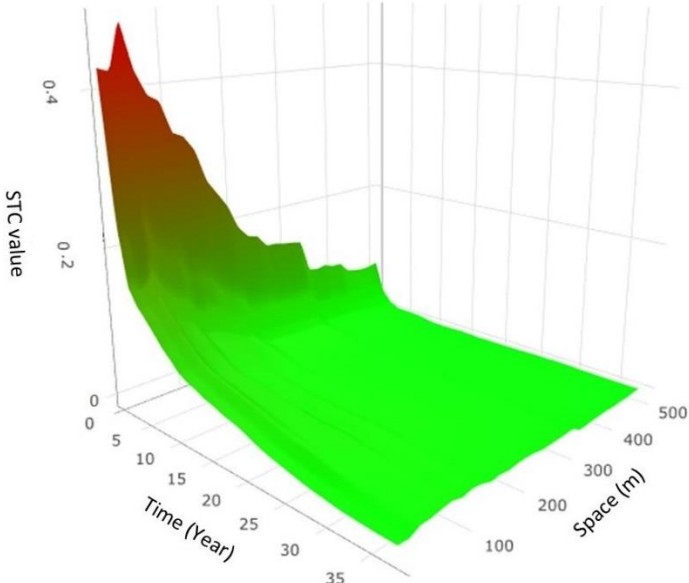

**Figure 6.** Space-time dynamic of landslide path dependency. The colours represent the intensity of STC measure. Red color indicates high STC and green indicates low STC.

The exponential decay function that was fitted to the empirical STC values is:

$$Smoothed\ STC(t,d) = 0.44 * e^{(-t/16.7)} * e^{(-d/58.8)} \qquad (7)$$

This function shows that the STC measure decays exponentially over a characteristic time scale of 16.7 years and characteristic spatial scale of 58.8 meters. The residual standard error of the exponential function is 0.01, in units of STC (-), which compares favourably with the actual values that range up to 0.44.

### 4.2 Model performance

We compared performance of the conventional, conventional plus path dependent, and purely path dependent landslide susceptibility models, using AUC (greater is better) and AIC (lower is better) values as measure of performance. The best performing landslide susceptibility model was the conventional plus path dependent model, both when expressed as AUC values and as AIC values (Table 1). The purely path dependent landslide susceptibility model, constructed with only the two landslide path dependency variables, performed better than the conventional landslide susceptibility model with its 16 DEM-derived variables.

**Table 1.** Performance of the three landslide susceptibility models. The values of AUC represent the average AUC
values in the 10 training and 10 testing datasets. The values of AIC represent the average AIC values in the 10
training datasets.

| AUC and AIC values | Conventional susceptibility model | Conventional plus path dependent susceptibility model | Path dependent susceptibility model |
|---|---|---|---|
| AUC training | 0.704 ± 0.006 | 0.764 ± 0.003 | 0.721 ± 0.004 |
| AIC training | 12,678 ± 82 | 11,711 ± 53 | 12,469 ± 62 |
| AUC testing | 0.682 ± 0.007 | 0.732 ± 0.004 | 0.698 ± 0.004 |



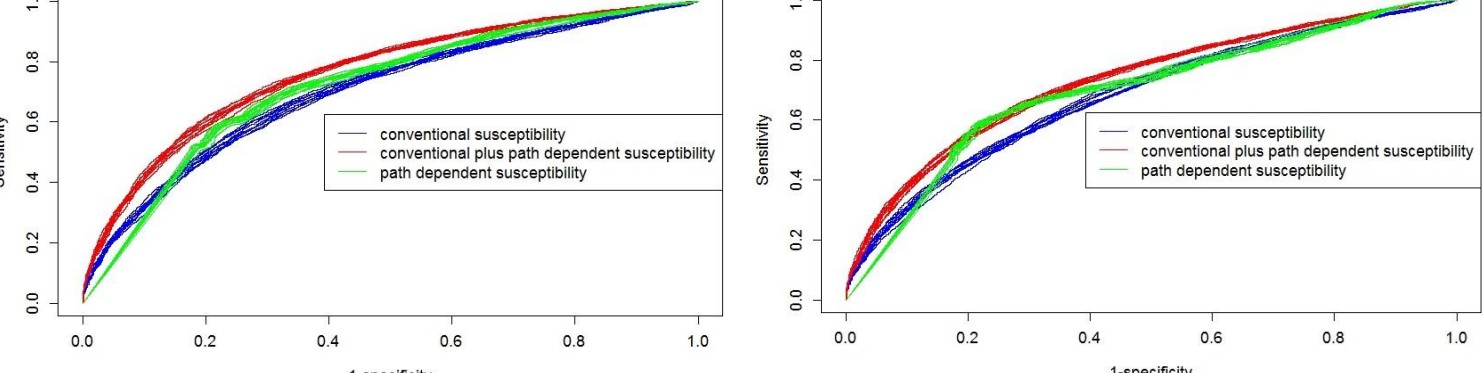

**Figure 7.** Receiver operating characteristic (ROC) curves of the three landslide susceptibility models in the 10
training datasets (left) and in the 10 testing datasets.
For conventional susceptibility models, 6 DEM-derivatives were selected in all 10 models (Table 2). Adding two
landslide path dependency variables into DEM-derivatives variables affected the inclusion and exclusion of DEM-
derivative variables only slightly. For example, the variables TPI and distance to river were selected 4 and 7 times
respectively in the conventional susceptibility models whereas after adding the two landslide path dependency
variables, these variables were selected 5 and 4 times respectively. The variable eastness which was selected twice
in the conventional susceptibility models, was never selected in the conventional plus path dependent susceptibility
models.






**Table 2.** Selection of independent variables in conventional, conventional plus path dependent and purely path
dependent landslide susceptibility modelling. Variables selected 6 or more times are shown. The numbers between
parentheses indicate how often variables were selected.

| Three landslide susceptibility models | Number of variables selection in 10 times repetition | Average number of variables selected in the three susceptibility models |
|---|---|---|
| Conventional (16 DEM-derivatives) | Elevation (10), standard deviation of slope (10), LS factor (10), standard deviation of elevation (10), stream power index (10), aspect (10), distance to river (7), vertical distance to channel network (6), relative slope position (6) | 8.7 |
| Conventional plus path dependent (16 DEM-derivatives plus two landslide path dependency variables) | Elevation (10), standard deviation of slope (10), LS factor (10), standard deviation of elevation (10), stream power index (10), aspect (10), max smoothed STC value (10), sum of all smoothed STC value (10) | 10.4 |
| Path dependent (two landslide path dependency variables) | max smoothed STC value (10), sum of all smoothed STC value (10) | 2 |


In all the training and the testing datasets, the contingency tables (Table 3) showed that conventional landslide
susceptibility models differed substantially from the conventional plus path dependent and path dependent
landslide susceptibility models. In particular, the percentage of false positives (the percentage of pixels without
landslides predicted with landslides) for the conventional susceptibility models is higher than for the two other
susceptibility models. However, there are also fewer true negatives (the percentage of pixels without landslides
predicted without landslides) in the conventional than in the conventional plus path dependent and path dependent
susceptibility models. The variation in the differences is larger in the training datasets than the testing datasets,
suggesting that all fitted models are robust.






**Table 3.** Contingency tables computed with cut off value of 0.5 for the three models. The numbers in the table
represent the average values computed in the 10 training and 10 testing datasets.

| | | Conventional landslide susceptibility | | Conventional plus path dependent landslide susceptibility | | Path dependent landslide susceptibility | |
|---|---|---|---|---|---|---|---|
| | | Observed landslides | | Observed landslides | | Observed landslides | |
| | | yes | no | yes | no | yes | no |
| Predicted landslides (training) | yes | 35 ± 0.33 | 19 ± 0.60 | 34 ± 0.42 | 14 ± 0.23 | 31 ± 0.8 | 13 ± 0.32 |
| | no | 15 ± 0.33 | 31 ± 0.60 | 16 ± 0.42 | 36 ± 0.23 | 19 ± 0.8 | 37 ± 0.32 |
| Predicted landslides (testing) | yes | 33 ± 0.50 | 19 ± 0.21 | 29 ± 0.35 | 13 ± 0.43 | 23 ± 0.24 | 12 ± 0.41 |
| | no | 17 ± 0.50 | 31 ± 0.21 | 21 ± 0.35 | 37 ± 0.43 | 27 ± 0.24 | 38 ± 0.41 |


### 3.3 Conventional, conventional plus path dependent and purely path dependent landslide susceptibility maps

The landslide susceptibility maps derived from the three models illustrate different patterns of landslide
susceptibility (Figure 8). For the models that include path dependency, the presented maps give the average values
of all simulated time slices. Differences between the maps correspond with the considerable differences in the
performance of their landslide susceptibility models in terms of AUC and AIC values (Table 1). The path
dependent landslide susceptibility map is visually different from both other landslide susceptibility maps, with the
pattern dominated by regions of high susceptibility around locations where landslides previously occurred.

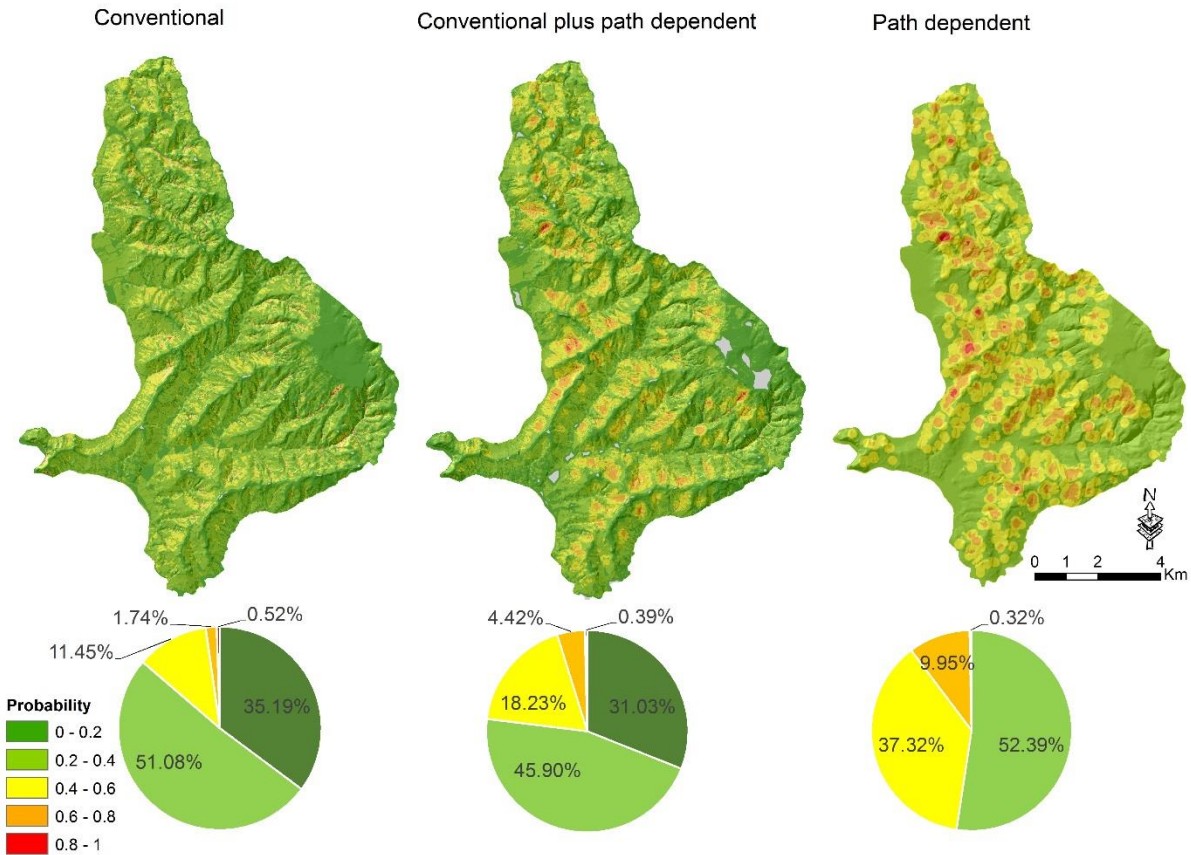

**Figure 8.** Conventional landslide susceptibility map in the left, the conventional plus path dependent landslide susceptibility map (averaged out over 16 time slices) in the middle and path dependent landslide susceptibility map (averaged out over 16 time slices) in the right. The pie charts show the percentage of pixels in each map in different probability levels of landslide occurrence.

The 16 conventional plus path dependent landslide susceptibility maps are dynamic and change over time (Figure 9). These changes reflect the exponential decay with increasing time since previous nearby landslides (Figure 6) and the sudden increase of susceptibility in areas close to recent landslides. The gradual decrease in susceptibility levels is clearest when comparing the 1981 and 2004 susceptibility maps, whereas the sudden increase is clearest when comparing the 2004 and 2014 maps. The 2014 susceptibility map has higher susceptibility levels because of the impact of recent landslides in the year 2013.

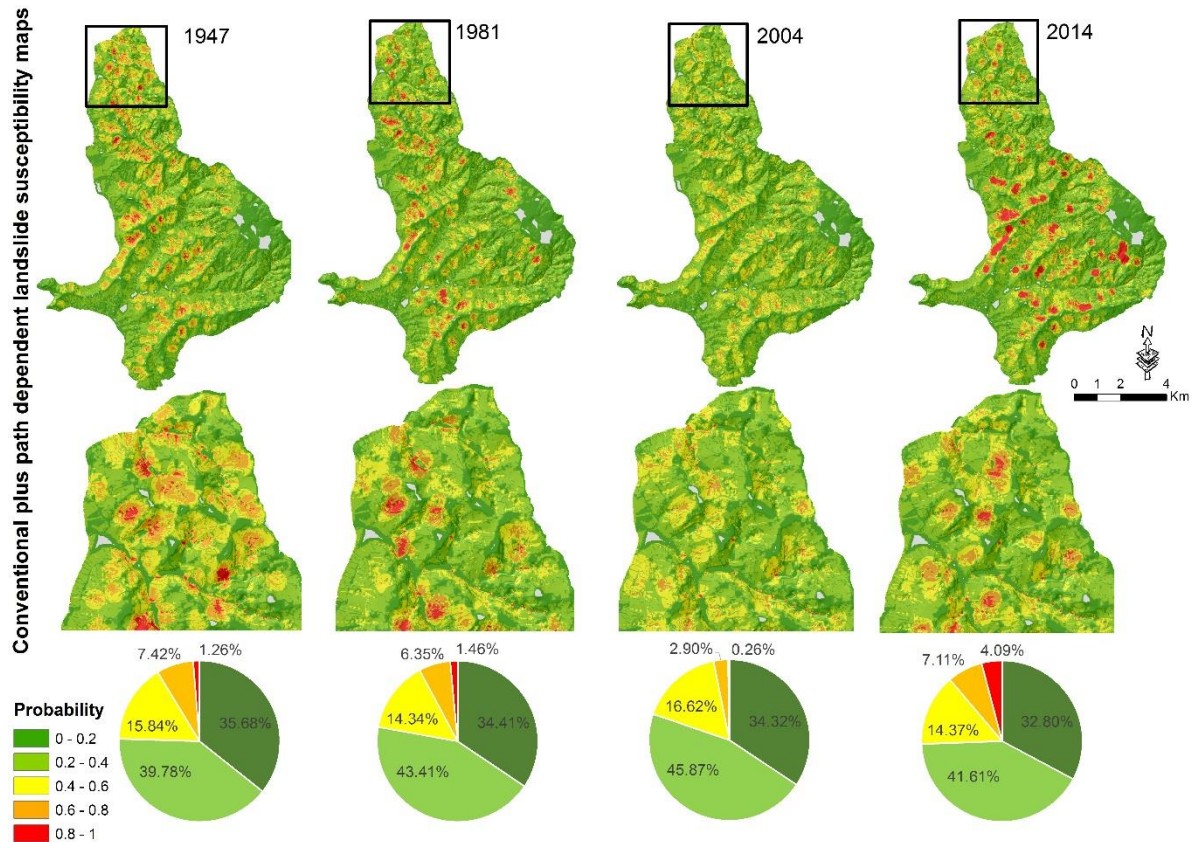

**Figure 9.** Examples of four dynamic conventional plus path dependent landslide susceptibility maps in the years of 1947, 1981, 2004 and 2014. Zoomed maps show the places where there are large changes in susceptibility over time.

Similar dynamics are visible when comparing landslide susceptibility maps constructed with the purely path dependent model for different years (Figure 10). These maps show only the pure influence of earlier landslides on susceptibility to future landslides (Figure 6). Again, the susceptibility of landslides decreases where distance from earlier landslides in space and time increases, but jumps back up when more recent landslides become part of the landslide history. The pure influence of each individual landslide on the susceptibility to the future landslide is strong when a landslide is fresh which is reflected in the high percentage of susceptibility levels of 0.6-0.8 and 0.8-1.0 in 1947 and 2014.When time passes since the previous landslide has occurred, the susceptibility decreases with an exponential decay response which is reflected in the low percentage of susceptibility levels of 0.6-0.8 and 0.8-1.0 in 1981 and 2004.

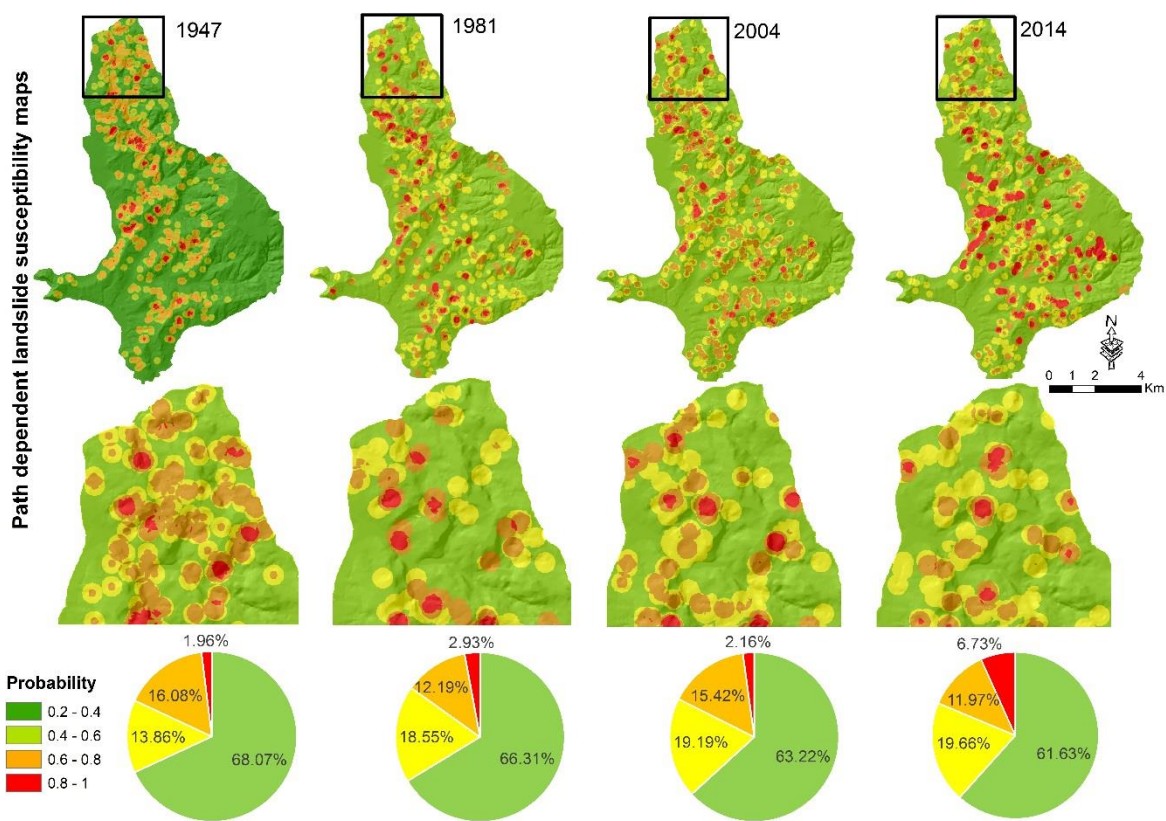

**Figure 10.** Examples of four dynamic path dependent landslide susceptibility maps in the years of 1947, 1981, 2004 and 2014. Zoomed maps show the places where there are large changes in susceptibility over time.

**5. Discussion**

In this section, we focus first on the quantification of landslide path dependency in the pixel-based multi-temporal landslide inventory, and then discuss its role in susceptibility models. We also discuss the susceptibility model performance for all three model types. At the end, the exportability of landslide path dependency parameters and the implication of dynamic time-variant path dependent landslide susceptibility in landslide hazard is discussed.

**5.1 Quantification of landslide path dependency**

The quantification of landslide path dependency using Ripley's space-time K function (Ripley, 1976; Diggle et al., 1995) indicates, in our study area, an exponential decay response in the STC values (Figure 6). This means that there is a positive influence of earlier nearby landslides on susceptibility that decays exponentially in time and space with a characteristic time scale of about 17 years, and a characteristic space scale of about 60 meters. This is in accordance with our previously quantified landslide path dependency using follow-up landslide fraction in which the decay period of landslide path dependency was found to be about two decades (Samia et al., 2017b). Landslide clustering manifests in the form of spatial association among landslides where follow-up landslides occur immediately after and close to a previous landslide (Samia et al., 2017a). Samia et al. (2017b) discussed the possible mechanism in the formation of clusters of landslides in which the size of the initial landslide and changes in hydrology of slope destabilized by a landslide could facilitate the occurrence of follow-up landslides and hence clusters of landslides.

STC values and their exponential decay to some extent depend on the method that we have chosen to determine
the centre point of landslides when converting polygons of landslides to points of landslides. Our approach was to
take the geometric centre, but other options exist (Haines, 1994) and their impact should be explored. Also, in the
computation of STC values with Ripley's space-time K function, distance between landslides was calculated using
the Pythagorean theorem without distinguishing between distances in the x and y direction. Also we did not include
differences in the elevation of centre points in our distance calculations. For future work, it could be interesting to
define one dimension as the distance along the slope in the downslope direction and another dimension as the
distance in the slope parallel direction, and keeping these two spatial dimensions separate in addition to the
temporal dimension.
**5.2 Effect of landslide path dependency on performance of landslide susceptibility models**
Our results demonstrated that including landslide path dependency effect in a pixel-based landslide susceptibility
model constructed by DEM-derivatives improves model performance substantially. This is in line with high AUC
and low AIC values for the conventional plus path dependent landslide susceptibility model (Table 1 and Figure
7). This confirms our main hypothesis that adding the effect of landslide path dependency boosts the performance
of landslide susceptibility models, and is in accordance with our previous expectations regarding stronger effect
of landslide path dependency in a pixel-based landslide susceptibility model than in a slope unit-based landslide
susceptibility model (Samia et al., 2018). Landslide path dependency is a local effect (apparently with
characteristic space scale of about 60 meters) in which an earlier landslide increases the likelihood of follow-up
landslide occurrence. Such a local effect is obviously more visible at pixel resolution of 10 m rather than at slope
unit resolution (with a median size of 51486 $m^2$ in our study area).
Strikingly, the purely path dependent landslide susceptibility model constructed with only the two landslide path
dependency variables performs better than the conventional landslide susceptibility model made by DEM-
derivative variables (Table 1 and Figure 7). This is potentially interesting since it implies that the landslide
inventory itself can be used to map susceptibility to landslide without using DEM-derivatives which have been
conventionally used in landslide susceptibility modelling (Varnes, 1984; Guzzetti et al., 2005). The performance
of this path dependency-only model thus highlights that proximity to previous landslides can adequately capture
susceptible locations. It also suggests that the path dependent models' success in our experiments may be partly
due to the fact that they capture static spatial effects that have not been resolved with our explanatory factors. It is
attractive to imagine follow-up work that attempts to disentangle this static spatial effect that is unrelated to
landslide history from dynamic spatial effects that are related to landslide history. The key to such disentangling
should be that the former does not decay over time, whereas the latter does. More advanced statistical approaches
that simultaneously estimate purely spatial and spatiotemporal effects may be needed. More complex explanatory
variables such as geology, soil and land use can also be used along with DEM-derivatives to improve landslide
susceptibility models and maps. However, these are not always available. In fact, considering landslide path
dependency effect into such complete explanatory factors improve their performance as well. We confirmed this
in an additional exploration where we constructed a conventional landslide susceptibility model used in this paper,
with the same DEM-derivatives, but also with land use and geology as explanatory factors. The results
demonstrated that adding our two landslide path dependency variables to such an improved conventional landslide
susceptibility increased its performance (from AUC value of 0.771 to AUC value of 0.801).

Another important aspect of considering landslide path dependency effect in landslide susceptibility modelling is providing dynamic landslide susceptibility maps. Landslide susceptibility maps are usually classified into five levels of probability to landslide occurrence ranging from 0 to 1. In the conventional landslide susceptibility map (Figure 8, left map), the five probability levels of susceptibility by definition remain constant over time since the DEM-derivatives in the model are constant (although DEM-derivatives also change when a landslide occurs, but DEMs are not updated frequently enough to reflect this). The usage of conventional static landslide susceptibility maps and dynamic landslide susceptibility maps taking landslide path dependency depends on the goal and task of audience. In reality, static susceptibility maps created (either with a conventional susceptibility model, or as the static portion of a conventional plus path dependent model) can be used in sustainable planning whereas dynamic susceptibility maps can be considered in short-term land use planning.

However, adding landslide path dependency in landslide susceptibility models, provides dynamic landslide susceptibility maps (Figures 9 and 10) in which the levels of susceptibility change over time, reflecting the exponential decay response of landslide path dependency (Figure 6). The changes are in the places where landslides have already occurred, mainly in probability levels of susceptibility ranging from 0.6 to 1.0. This suggests that the part of area located in the high probability level of susceptibility could switch to the low probability level of susceptibility (0 to 0.6) after a decade. This is exemplified between 1947 and 1954 landslide susceptibility maps, in which about 9 km$^2$ of study area drops more than 0.1 in their probability of landslide occurrence. After adding the two path dependency variables in the conventional landslide susceptibility modelled with DEM-derivatives, it turns out that the coefficients of all DEM-derivative variables become lower (e.g., LS factor becomes less important).

**5.3 Can landslide path dependency parameters be transported to other areas?**

In landslide prone areas where landslides are documented and mapped in the form of polygon-based multi-temporal inventories, the landslide path dependency can be quantified based on geographical overlap among landslides, and hence used in landslide susceptibility modelling (Samia et al., 2017b; Samia et al., 2018). However, polygon-based multi-temporal landslide inventories are rare to the best of our knowledge, and hence in many areas geographical overlap among landslides cannot be computed. In this paper, we proposed using Ripley's space-time K function to compute landslide path dependency where point-based multi-temporal landslide inventories are used. Using such inventories, our STC measure (Eq. 6) can be used to quantify path dependency among landslides.

It is attractive to think that the STC measure (Eq. 6) and its parameters (Eq. 7) can be directly exported to landslide prone areas with substantial geological and topographical similarities. However, to gain confidence in this approach, multi-temporal landslide inventories from such places (e.g., (Schlögel et al., 2011)) need to be interrogated to find out whether path dependency occurs, whether it occurs over similar space and time scales, and whether it adds value to susceptibility modelling. This would also allow us to start exploring what determines the characteristic space and time scales.

**5.4 Implications of path dependent landslide susceptibility in landslide hazard assessment**

We have already modified the definition of conventional landslide susceptibility modelling (Varnes, 1984;
Guzzetti et al., 2005) using spatial temporal dynamics of landslide path dependency (Samia et al., 2017a, b) as
following:
$Landslide\ susceptibility_{s,t} = f\big(conditioning\ attributes_s, landslide\ path\ dependency_{s,t}\big)$   (8)
In this study, both conventional plus path dependent and path dependent landslide susceptibility models turned out
to perform better than a conventional landslide susceptibility model (Table 1 and Figure 7). In both models,
availability of a space-time component – reflecting the exponential decay of landslide path dependency – indicates
that landslide susceptibility is dynamic. This challenges the way landslide hazard is assessed as landslide
susceptibility is an important element of landslide hazard.
In landslide hazard assessment, landslide susceptibility as a proxy of 'where landslides occur' is combined with
the temporal probability of landslide triggers (mainly rainfall) to determine 'when landslides occur' (Guzzetti et
al., 2006a). In this context, a dynamic landslide susceptibility (Eq. 8) needs to be considered in combination with
the temporal information of landslide triggers in the assessment of landslide hazard. When substantial landsliding
happens during a rainfall event, susceptibility in and around such landslides can be raised for a few decades in
which moderate rainfall events may already cause substantial landsliding, which raises susceptibility levels again.
(Figure 11). Such dynamics have been observed in a site near Seattle, Washington, where several new landslides
occurred in a slope that had recently experienced landslide activity whereas a nearby hillslope with the same
characteristics but without recently landslide activity did not experience new landsliding (Mirus et al., 2017). If no
substantial triggering event happens over the characteristic time scale of roughly 17 years, the increased
susceptibility will be substantially reduced, and a later rainfall event may have less influence on landsliding; the
probability of experiencing a follow-up landslide will have decreased.

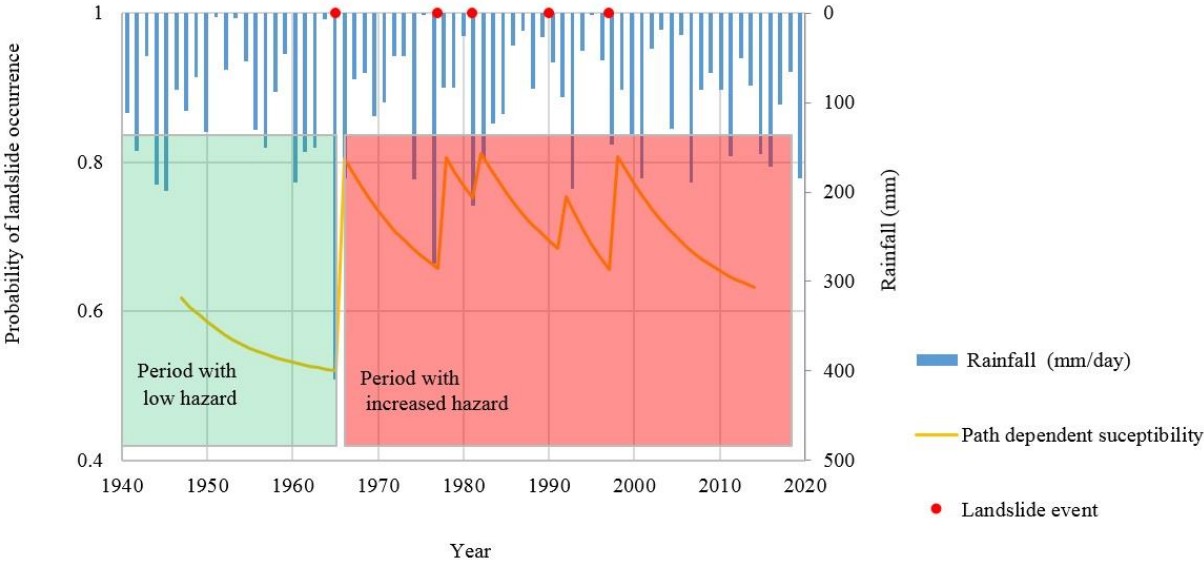

**Figure 11.** Conceptual model of the implication of dynamic path dependent landslide susceptibility model in
landslide hazard assessment. When susceptibility is low, the hazard is also low (providing the other components
of landslide hazard e.g., size remain unchanged) and large rainfall events are needed to trigger new landslides.
Then, when susceptibility is raised by such landslides, the hazard is also high and small rainfall events may trigger
new landslides.
6. **Conclusion**
In the Collazzone study area, in Central Italy, quantification of landslide path dependency reveals an exponential
decay response in landslide susceptibility as a function of space and time distance to earlier nearby landslides. For
our study area, the characteristic time scale of this effect is about 17 years and the characteristic space scale is
about 60 meters. Adding such an exponential decay response of landslide path dependency in conventional pixel-
based landslide susceptibility modelled by DEM-derivative improves the performance of model substantially.
Taking into account landslide path dependency effects in landslide susceptibility results in dynamic landslide
susceptibility models where susceptibility changes over time. We stress that landslide susceptibility modelling
should take the effect of landslide path dependency into account since it provides an estimation of temporal
validation of different probability levels of landslide occurrence in landslide susceptibility map. The obtained
landslide path dependency parameters can possibly be used for dynamic landslide susceptibility modelling in
landslide prone areas with environmental and data similarities. We proposed a conceptual model that considers
the impact of dynamic path dependent landslide susceptibility on landslide hazard.

Code and data availability. The multi-temporal landslide inventory, DEM, Geological and land use data of
Collazzone study area in Italy, was provided by CNR IRPI and can be requested from FG and FA. The code for
analysis of Ripley's space-time K function can be requested from JS and AT.
Author contributions. JS formulated the objective, designed the methodology and conducted the GIS, statistical,
space-time Ripley's analysis and modelling of landslide susceptibility. AT provided methodological support,
advise and also made major revisions to the paper. AB and JW made major revisions to the paper. FG and FA
provided the data and also made major revisions to the paper.
Competing interests. The authors declare that they have no conflict of interest.
Acknowledgement. This research is part of J Samia PhD project at Wageningen University and Research, funded
by Ministry of Science, Research and Technology of Iran, Laboratory of Geo-Information Science and Remote
Sensing and Soil Geography & Landscape group of Wageningen University, and supported by the Geography
Department of Kansas State University in USA. The authors would like to thank Dr Ben Mirus, the anonymous
reviewer, and also the editor Professor Thomas Glade for their constructive comments and valuable suggestions,
which have greatly improved the quality of the paper.

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
