# Peer review of "Dynamic path dependent landslide susceptibility modelling"

_Natural Hazards and Earth System Sciences, 2019_

## Referee Comment (RC1) · Ben Mirus (Referee) · 31 Jul 2019

Summary and Assessment:

The authors present an interesting study that builds on their prior work on the "path-dependency" phenomena in landslide occurrence. While the current and previous studies rely on the same multi-temporal landslide inventory from the Collazzone study area, Italy, the current study examines a finer spatial resolution (10x10m as opposed to individual watersheds) and also uses a new metric to quantify the spatio-temporal correlation component (Ripley's K coefficient). The study uses an appropriate split sampling approach to calibrate and validate the different susceptibility models as well as the widely used AUC metric from ROC analysis. This new approach using estab-

lished techniques provides a different result than the prior coarser application, which ultimately supports the importance of time-variable susceptibility models for this site and likely other landslide sites. The results further indicate that while the most complex "conventional plus path dependent" model exhibits the best performance, the "path dependent only" performs better than the "conventional' model despite having far fewer parameters.

Overall, the topic will be of considerable interest to readers of NHESS and the novel methods and contribution are suitable for publication. The manuscript is well written and logically organized. The figures are clear, and referencing is appropriate. I have only a few concrete suggestions for improvement and one general comment. Relatively trivial revisions should be sufficient to address these.

General Comment:

Given the superior performance of the "path dependent only" model over the "conventional" model, it is important to note that at least the spatial part of the path dependent model may not be entirely related to the occurrence of past landslides. Rather this might also be at least partially explained by the generally accepted phenomenon that landslides tend to happen where they have already occurred previously. That is, there could be factors that are not considered in the conventional model that explain the "where" of landslide occurrence better than the terrain attributes considered. For example, soil thickness or hydromechanical properties or climatic and environmental forcing factors that vary independently of topography.

Thus, it is unclear how much the timing of previous landslides is relevant compared to merely the occurrence of past landslides. Perhaps this would be beyond the scope of this study, but it would be very interesting if the authors could somehow separate the spatial and temporal element to see how much of the model improvement is related to "landslides occur here" vs. "a landslide just occurred here" phenomena. This is interesting because it offers the possibility that historic landsliding (without a multi-temporal

dataset) could be used as a variable to improve susceptibility modeling instead of trying to account for the difficult to measure subsurface variables.

These points are worth mentioning in the discussion section to provide further context for the significance of the contribution and possible future directions.

Specific Suggestions:

L214. Here and elsewhere I suggest using the Oxford comma to appropriately distinguish between items in a list. It is used sometimes in the manuscript but not consistently.

L225. Figure 6. What does the color red/green represent?

L270. Figure 7. In the Path dependent only case, there is clearly some red (0.8-1.0) on the map, but the pie chart indicates that 0% are in this class?

L375. There are three models considered, so perhaps edit to: "In both improved models..."

L381-387. It could be worth mentioning that precisely this phenomenon was observed at our site near Seattle (Mirus et al., 2017), namely for the same sequence of storm events a previous landslide remobilized multiple times whereas a neighboring hillslope with the same terrain attributes did not fail.

---

## Referee Comment (RC2) · Anonymous Referee #2 · 6 Sep 2019

Summary

In this manuscript the authors present a landslide susceptibility analysis for a study area in Italy based on a multi-temporal landslide inventory employing logistic regression. They introduce two new spatial variables to account for past landslides and conclude that based on Area Under the Curve and Akaike Information Criterion metrics models considering the variables reflecting the landslide history perform better than a more conventional model based on digital elevation model (DEM) derived variables.

General remarks

The paper is written in good English and it is well structured. The concept of path dependency is introducing a new idea to landslide susceptibility mapping, and it is

interesting to see that this phenomenon manifests in the data.

It should be noted though that this is the fourth publication of the authors with the same dataset for a small study area. I know that such multi-temporal datasets are scarce, but in order to prove the idea I think it is also important to test the concept with other datasets rather than introducing yet another tweak in the methodology.

It is also hard for me to follow the interpretation and conclusion that the models considering path dependency are "substantially" better than the conventional one. With the information provided I consider it hard to take a decision at all, but I would tend to rate the conventional model as the best, see also my comments below.

I think it is necessary to also show the detailed susceptibility maps of the conventional model to be able to compare the spatial performance. In the end, it is important that a model is making sense in the spatial context to see if it is plausible and useful on an operational level. In all path dependent susceptibility maps it is obvious that the path dependent variables clearly dominate the spatial distribution of landslide susceptibility (bullseye artifacts). Are those "hot spot" susceptibility maps useful in practice? Are the models well-balanced? The conventional model probably has a poor variability as it only contains DEM derived variables. What happens if more fundamental information is introduced, like lithology? I understand that it was intended to use a model with minimal data requirements, but it also hast to be demonstrated first that this works comparing it to a more complex dataset.

The methodology is also not completely clear to me based on the explanations provided. Were models produced for different time slices or only one model for each parameter set? See also comments below.

To sum it up, although I believe the idea of the authors to introduce landslide history in susceptibility modelling is interesting and the paper is well written, I do not agree with the main conclusions based on the information provided. Thus, in my opinion more information and a more critical discussion are required for publication. Please

find some more detailed comments and suggestions below.

Specific comments

Methodology:

E.g. L 94-98: To better understand the whole concept it would be good to understand how the authors define "follow-up landslides" and if/how they are for example discriminated from reactivated landslides.

L 112-113, Fig. 1: Is this figure not taken from Samia et al. 2017a or b?

L 183, Fig. 5: The figure is not referred to in the text. Is it correct that the arrows on the left point from the start to the results? Are they not supposed to start at the Smoothed STC sketch?

L 205-215: I am not sure if I understand the composition of the training and testing data and the whole procedure. Were the models trained on a single time slice from L 201 each and then tested with the subsequent testing time slice from L 202? Then 10 samples were taken for each time slice? How were the results in tables 1 and 3 generated from the different time slice models? Maybe the methods section could be put more clearly.

Results and discussion:

L 218-219: Is it possible to show a map of what the variables reflecting the path dependency look like spatially?

L 225, Fig. 6: What does the color code represent?

L 228-230: Isn't a spatial scale of 60 m quite small? Because 60 m can be below the size of a single landslide. Are these new landslides or reactivated ones?

Table 1, Table 3: Are the results available for different time slices? It is unclear to me which results are presented here. Is this a summary of all time slice models? Or the

best models?

Section 5.2, Table 3: I do not agree with the interpretation that the conventional plus path dependent and path dependent models are substantially better than the conventional one. The conventional one has slightly more hits and less misses (false negative). In my opinion, false negatives are more critical than false positives, which actually contribute to a better zonation when the goal is not the accurate detection of landslides but the identification of susceptible areas. It would be interesting to see also the susceptibility maps (like figures 8 and 9) for the conventional models to be able to better compare their spatial performance. Success rate curves plotting the distribution of landslides over the susceptibility classes would add more information.

L 235-237 and L 339-341, using only path dependent variables: I do not understand, why should we want to predict landslides just based on past landslides? Firstly, at this point multi-temporal landslide inventories are rarely available and secondly, this is in my opinion in disagreement with the fundamental paradigm of data-driven landslide susceptibility analysis, which is deducing landslide occurrence from independent variables. Also, the susceptibility maps based on the path dependent variables only have extreme bullseye effect artifacts and I doubt that the maps are in this form useful for practical implementations.

Figures 7, 8 and 9:

- the maps would be easier to interpret with a hillshade in the background and the outlines of the corresponding training and/or test landslides, which are required to assess the spatial performance of the models.

- it is a good idea to show the distribution of the susceptibility classes, but pie charts are not very effective for comparing multiple part-to-whole relationships. They are inconvenient to read and it is hard to perceive the quantitative relationships. Bar or column charts would be more suitable.

- why are there blank/white areas in the maps containing path dependent variables?

L 345: I think this should be the map on the left in Figure 7.

L 347-349, usage of landslide susceptibility maps for amount of time of landslide inventory: I think this is hard to generalize and depends on the task, but for sustainable planning of resilient urban areas I would rather counsel time-insensitive susceptibility models based on intrinsic parameters. Figure 10: I do not understand what hypothetical means. Is this graph based on real data or is this just a sketch?

---

## Author Comment (AC2) · 5 Nov 2019

Dear Reviewer,

Thank you for your nice and constructive comments. We have addressed your comments point-by-point and provided the answer, explanation, and modification where are required in the revised manuscript. This has been uploaded as a supplement file where you can find our answers in a table plus the revised manuscript with track changes.

We hope our answers satisfy your demands.

Best regards, Jalal Samia and the co-authors

Please also note the supplement to this comment:

[Figure]

https://www.nat-hazards-earth-syst-sci-discuss.net/nhess-2019-125/nhess-2019-125-AC2-supplement.pdf

---

## Author Response (AR1)

Dear Dr Ben Mirus,

Thank you for your nice and constructive comments. We have addressed your comments point-by-point and provided the answer, explanation, and modification where are required in the revised manuscript. This has been uploaded as a supplement file where you can find our answers in a table plus the revised manuscript with track changes.

We hope our answers satisfy your demands.

Best regards,

Jalal Samia and  the co-authors

Referee's comments:                                            Authors responses :

| Comment | Response |
|---|---|
| General Comment:

Given the superior performance of the "path dependent only" model over the "conventional" model, it is important to note that at least the spatial part of the path dependent model may not be entirely related to the occurrence of past landslides. Rather this might also be at least partially explained by the generally accepted phenomenon that landslides tend to happen where they have already occurred previously. That is, there could be factors that are not considered in the conventional model that explain the "where" of landslide occurrence better than the terrain attributes considered. For example, soil thickness or hydromechanical properties or climatic and environmental forcing factors that vary independently of topography. Thus, it is unclear how much the timing of previous landslides is relevant compared to merely the occurrence of past landslides. Perhaps this would be beyond the scope of this study, but it would be very interesting if the authors could somehow separate the spatial and temporal element to see how much of the model improvement is related to "landslides occur here" vs. "a landslide just occurred here" phenomena. This is interesting because it offers the possibility that historic landsliding (without a multi-temporal dataset) could be used as a variable to improve susceptibility modeling instead of trying to account for the difficult to measure subsurface variables. These points are worth mentioning in the discussion section to provide further context for the significance of the contribution and possible future directions. | Thank you for such a nice remark. Indeed this might be that the spatial part of path dependent model is not entirely related to the space-time effect of past landslides but could be partly because those places are unstable slopes where landslides keep occurring. The two landslide path dependency variables implemented in the conventional plus path dependent and purely path dependent landslide susceptibility models were derived simultaneously from Ripley's space-time k function in the line of joint spatial and temporal effect of landslide path dependency on future susceptibility, and are not separable with Ripley's function.

In the revised manuscript, we have considered your point in the discussion as following:

"The performance of this path dependency-only model thus highlights that proximity to previous landslides can adequately capture susceptible locations. It also suggests that the path dependent models' success in our experiments may be partly due to the fact that they capture static spatial effects that have not been resolved with our explanatory factors. It is attractive to imagine follow-up work that attempts to disentangle this static spatial effect that is unrelated to landslide history from dynamic spatial effects that are related to landslide history. The key to such disentangling should be that the former does not decay over time, whereas the latter does. More advanced statistical approaches that simultaneously estimate purely spatial and spatiotemporal effects may be needed." |
| Specific Suggestions:

L214. Here and elsewhere I suggest using the Oxford comma to appropriately distinguish between items in a list. It is used sometimes in the manuscript but not consistently. | We took care of this throughout the manuscript in the updated version. |
| Specific Suggestions:

L225. Figure 6. What does the color red/green represent? | The colours represent the intensity of STC measure.
We have added two sentences to the caption of figure 6 to make this clear as following:

"The colours represent the intensity of STC measure. Red colour indicates high STC and green indicates low STC." |
| Specific Suggestions: | Thank you for your observation. Indeed this is not immediately clear on the pie chart of only path |

| | |
|---|---|
| L270. Figure 7. In the Path dependent only case, there is clearly some red (0.8-1.0) on the map, but the pie chart indicates that 0% are in this class? | dependent map. The 0 % belongs to the probability class of 0-0.2 where is not computed in the only path dependent susceptibility map nor. The 0.32% belongs to the red colour in the map. We removed 0% and updated the map in the revised manuscript, and now the number is clear on the pie chart. |
| Specific Suggestions:

L375. There are three models considered, so perhaps edit to: "In both improved models. . ." | Thank you for your suggestion, indeed that works better. We have edited that sentence to your suggested one. |
| Specific Suggestions:

L381-387. It could be worth mentioning that precisely this phenomenon was observed at our site near Seattle (Mirus et al., 2017), namely for the same sequence of storm events a previous landslide remobilized multiple times whereas a neighboring hillslope with the same terrain attributes did not fail. | Indeed an example of observed dynamic susceptibility in your study area could be mentioned in that section of manuscript. We have added this point as following in the updated manuscript:

"Such dynamics have been observed in a site near Seattle, Washington, where several new landslides occurred in a slope that had recently experienced landslide activity whereas a nearby hillslope with the same characteristics but without recently landslide activity did not experience new landsliding (Mirus et al., 2017)." |

Dear Reviewer,

Thank you for your nice and constructive comments. We have addressed your comments point-by-point and provided the answer, explanation, and modification where are required in the revised manuscript. This has been uploaded as a supplement file where you can find our answers in a table plus the revised manuscript with track changes.

We hope our answers satisfy your demands.

Best regards,

Jalal Samia and  the co-authors

Referees' comments:                                          Author's responses :

| Comment | Response |
|---|---|
| General remarks

1. It should be noted though that this is the fourth publication of the authors with the same dataset for a small study area. I know that such multi-temporal datasets are scarce, but in order to prove the idea I think it is also important to test the concept with other datasets rather than introducing yet another tweak in the methodology. | Thank you for this remark. This is absolutely true, and indeed the concept of landslide path dependency should be explored in other landslides prone areas providing that multi-temporal landslide inventories are available. We have two arguments for this:

a) As you mentioned, in order to explore the effect of landslide path dependency, the main requirement is the availability of multi-temporal landslide inventory which to the best of our knowledge (at the starting time of this project, start of PhD of the corresponding author in 2013) was rare. We are now aware of a few more multi-temporal inventories (Asturias in Spain and Ubaye Valley in France), that we are going to use then in a follow-up project if the proposal of that is granted.

b) The reason for another tweak in the methodology is that in the implementation of landslide path dependency on landslide susceptibility model in the resolution of slope unit (see Samia et.al, 2018), we found that the performance of landslide susceptibility model did not change substantially. We had this reasoning that the effect of landslide path dependency might be captured better in more finer resolution in the mapping unit of landslide susceptibility model due to the local effect of landslide path dependency. The results of this paper confirmed our previous reasoning. Therefore, this justifies our another tweak in the methodology where we converted the polygons of landslides to point, and then used a new metric (Ripley function) to quantify the effect of landslide path dependency. Having said, and considering the substantial importance of landslide path dependency on the performance of landslide susceptibility model, we believe that landslide path dependency is an important component in landslide susceptibility definition, and this should be further explored in other landslide prone areas. |
| General remarks

2. It is also hard for me to follow the interpretation and conclusion that the models considering path dependency are "substantially" better than the conventional one. With the information provided I consider it hard to take a decision at all, but I would tend to rate the conventional model as the best, see also my comments below. | The conclusion comes from the comparison of performance of three landslide susceptibility models with the metrics of AUC and AIC. This improvement comes from adding only two variables reflecting landslide path dependency to the existing 16 DEM-derivatives. An increase from the AUC of conventional susceptibility model with the value of 0.704 to the AUC value of 0.764 for the conventional plus path dependent susceptibility refers to "substantial" improvement of landslide susceptibility model. Also, AIC values for conventional plus path dependent and purely path dependent landslide susceptibility models are lower |

| | (11711 and 12469) which are lower than the AIC value of conventional susceptibility model (12678). |
|---|---|
| General remarks

3. I think it is necessary to also show the detailed susceptibility maps of the conventional model to be able to compare the spatial performance. In all path dependent susceptibility maps it is obvious that the path dependent variables clearly dominate the spatial distribution of landslide susceptibility (bullseye artifacts). Are those "hot spot" susceptibility maps useful in practice? Are the models well-balanced? | Regarding the detailed maps of conventional susceptibility see our answers in the comment number 14 in the below.

we have considered your comment regarding the applicability of dynamic landslide susceptibility maps in practice as following:

"In reality, static susceptibility maps created (either with a conventional susceptibility model, or as the static portion of a conventional plus path dependent model) can be used in sustainable planning whereas dynamic susceptibility maps can be considered in short-term land use planning."

The hotspots in the susceptibility maps of conventional plus path dependent and purely path dependent landslide susceptibility, show the predicted susceptibility per time slice given the history effect of previous landslides. Surely those hotspots are useful in practice in that given period as it reflects that newly happened landslides have higher susceptibility level demonstrating more cares for practical purposes.

We are not quite sure what you mean with well-balanced model but our datasets used in the three susceptibility models are balanced. |
| General remarks

4. The conventional model probably has a poor variability as it only contains DEM derived variables. What happens if more fundamental information is introduced, like lithology? I understand that it was intended to use a model with minimal data requirements, but it also hast to be demonstrated first that this works comparing it to a more complex dataset. | Thank you for this point. Aa you also have mentioned, this was our aim to model landslide susceptibility using minimum data requirements (DEM-derivatives) plus variables derived from landslide path dependency. However, we did that work and it's just not in the paper. The conventional susceptibility modelled by DEM-derivatives, geology and land use has lower model performance (AUC = 0.771) when adding two landslide path dependency variables in the conventional plus path dependent landslide susceptibility (AUC = 0.801).

Considering your comment, we also feel that this could be mentioned in the discussion of paper as following:

"More complex explanatory variables such as geology, soil and land use can also be used along with DEM-derivatives to improve landslide susceptibility models and maps. However, these are not always available. In fact, considering landslide path dependency effect into such complete explanatory factors improve their performance as well. We confirmed this in an additional |

| | exploration where we constructed a conventional landslide susceptibility model used in this paper, with the same DEM-derivatives, but also with land use and geology as explanatory factors. The results demonstrated that adding our two landslide path dependency variables to such an improved conventional landslide susceptibility increased its performance (from AUC value of 0.771 to AUC value of 0.801)." |
|---|---|
| General remarks

5. The methodology is also not completely clear to me based on the explanations provided. Were models produced for different time slices or only one model for each parameter set? | Indeed the methodology for modelling part is not completely clear. We have updated the manuscript in this matter (in line 216-220) as following:

"Conventional landslide susceptibility was modelled using DEM-derivatives only once for the defined training dataset and was tested using the independent testing dataset. Conventional plus path dependent landslide susceptibility model was constructed using DEM-derivatives plus the two landslide path dependency variables. The purely path dependent landslide susceptibility was modelled only by using the two landslide path dependency variables. All three models were constructed only once." |
| Methodology

6. E.g. L 94-98: To better understand the whole concept it would be good to understand how the authors define "follow-up landslides" and if/how they are for example discriminated from reactivated landslides. | This comment is confusing since in the lines of 94-98, we did not talk about follow-up landslides at all. However to make it clear, in our previous paper in Samia et al, 2017, we introduced for the first time the term "follow-up landslides" and differentiated that from reactivated landslides as following:

"Note that follow-up landslides are not reactivated landslides. We consider a landslide a reactivated landslide when all or most of the landslide moved down again, under the same general condition as the first landslide. Instead, follow-up landslides are new landslides that have different size and shape than the pre-existing landslide."

In the current manuscript when talking about follow-up landslides (mainly in the discussion), the proper reference has been provided. |
| Methodology

7. L 112-113, Fig. 1: Is this figure not taken from Samia et al. 2017a or b? | Yes, it is taken from those papers, and also from our paper in Samia et al, 2018. We have added those two references in the updated manuscript. |

| | |
|---|---|
| Methodology

8. L 183, Fig. 5: The figure is not referred to in the text. Is it correct that the arrows on the left point from the start to the results? Are they not supposed to start at the Smoothed STC sketch? | Thank you for this point, indeed the figure is not referred in the text. Now we have provided reference for this figure in the line of 175 where we talk about the computation of the two landslide path dependency variables.

Also we appreciate your comment about the position of arrows in Fig. 5. You are right and we corrected the figure in the updated manuscript. |
| Methodology

9. L 205-215: I am not sure if I understand the composition of the training and testing data and the whole procedure. Were the models trained on a single time slice from L 201 each and then tested with the subsequent testing time slice from L 202? Then 10 samples were taken for each time slice? How were the results in tables 1 and 3 generated from the different time slice models? Maybe the methods section could be put more clearly. | No, the model was trained to the combination of time slices 1947, 1954, 1981, 1985, 1999, May 2004, March and May 2010. After that, the model was tested on testing dataset which is the combination of time slices of 1965, 1977, 1991, 1997, December 2004 and 2005 and April 2013 and 2014. Then, due to the unequal amount of pixels with and without landslide in each of these training and testing datasets, we selected 5000 pixels with landslides and 5000 pixels without landslides randomly in each of training and testing dataset. This random selection was repeated 10 times in both training and testing datasets. Therefore, for each of these 10 random datasets, conventional, conventional plus path dependent and purely path dependent landslide susceptibility models were constructed, and the results presented in the table 1 and table 3, are the average of these 10 models.

To make this part more clear, we have modified sentences in the lines of 200-203 as following:

"To achieve this, all landslides in the time slices of 1947, 1954, 1981, 1985, 1999, May 2004, March and May 2010 were used for training, and all landslides in the time slices of 1965, 1977, 1991, 1997, December 2004 and 2005 and April 2013 and 2014 were used for testing (Figure 1)."

Also sentences in the lines of 209-213 were updated as:

"Therefore, we randomly selected 5,000 pixels with landslides and 5,000 pixels without landslides from both training and testing datasets in order to create equal datasets both for training and testing of the models. This random selection of pixels was repeated 10 times both in the training and testing datasets. Therefore, we trained the conventional, conventional plus path dependent and purely path dependent landslide susceptibility 10 times, and finally tested 10 times as well." |

| | |
|---|---|
| Results and discussion

10. L 218-219: Is it possible to show a map of what the variables reflecting the path dependency look like spatially? | No that's not possible. We can only make prediction from these variables depending when and where previous landslides happened. We have already shown the examples of the predictions from landslide path dependency variables in our existing figures 8 and 9. |
| Results and discussion

11. L 225, Fig. 6: What does the color code represent? | The colors represent the intensity of STC measure.
We have added two sentences to the caption of figure 6 to make this clear as following:

"The colours represent the intensity of STC measure. Red colour indicates high STC and green indicates low STC." |
| Results and discussion

12. L 228-230: Isn't a spatial scale of 60 m quite small? Because 60 m can be below the size of a single landslide. Are these new landslides or reactivated ones? | We noted that the 60 meters is the characteristic spatial scale of the exponential function so that substantial effects still exist over distances more than hundred meters. Moreover, this has been calculated from centre point to the centre point of different landslides and so if the centre points are 60 or 100 or 150 meters away from each other, they could still be not overlapping or overlapping and they could be either new or reactivating landslide. This empirical procedure does not distinguish between new and reactivated landslides. |
| Results and discussion

13. Table 1, Table 3: Are the results available for different time slices? It is unclear to me which results are presented here. Is this a summary of all time slice models? Or the best models? | Considering the explanations given above in the methodology part, the results presented in these two tables should already be clear. However, to make this even more clear, we have modified the caption of table 1 and table 3 as following:

"The values of AUC represent the average AUC values in the 10 training and 10 testing datasets. The values of AIC represent the average AIC values in the 10 training datasets."

"Contingency tables computed with cut off value of 0.5 for the three models. The numbers in the table represent the average values computed in the 10 training and 10 testing datasets." |
| Results and discussion | We have explained this in above. Just to recall, this conclusion comes from the comparison of AUC and AIC values of these three models and not based on the |

| | |
|---|---|
| 14. Section 5.2, Table 3: I do not agree with the interpretation that the conventional plus path dependent and path dependent models are substantially better than the conventional one. The conventional one has slightly more hits and less misses (false negative). In my opinion, false negatives are more critical than false positives, which actually contribute to a better zonation when the goal is not the accurate detection of landslides but the identification of susceptible areas. It would be interesting to see also the susceptibility maps (like figures 8 and 9) for the conventional models to be able to better compare their spatial performance. Success rate curves plotting the distribution of landslides over the susceptibility classes would add more information. | values presented in table 3, which result only from an arbitrary 0.5 cut off value.

It's not possible to make conventional susceptibility map like the dynamic path dependent susceptibility maps in figure 8 and 9 because only one conventional landslide susceptibility map was made based on the conventional definition of landslide susceptibility which is a time-invariant concept. The relevant spatial comparison is shown in figure 7 where we show the static susceptibility map from conventional model along with examples from one time slice of the other two path dependent dynamic models.

We have provided the success rate curves for three models in 10 training datasets and also for the testing datasets as new figure 7 in the revised manuscript. |
| Results and discussion

15. L 235-237 and L 339-341, using only path dependent variables: I do not understand, why should we want to predict landslides just based on past landslides? Firstly, at this point multi-temporal landslide inventories are rarely available and secondly, this is in my opinion in disagreement with the fundamental paradigm of data-driven landslide susceptibility analysis, which is deducing landslide occurrence from independent variables. Also, the susceptibility maps based on the path dependent variables only have extreme bullseye effect artifacts and I doubt that the maps are in this form useful for practical implementations. | Thank you for this interesting question and comment. Well, we – for the first time – have challenged the conventional definition of landslide susceptibility with introducing the new concept of path dependency indicating the history effect of landslides on future susceptibility (Samia et al, 2017). With this, landslide susceptibility is not a function that considers only the spatial distribution of landslides along with a set of independent environmental factors but path dependency also needs to be taken into account. This effect was found in exploration of a unique and rich multi-temporal landslide inventory in Collazzone, Italy. Landslide path dependency in this study area indicated that susceptibility is not time-invariant but susceptibility changes over time. With this, a new paradigm for landslide susceptibility is emerging called dynamic path dependent susceptibility which requires exploration of the existence of path dependency, characterization and quantification of such effect, and finally its implementation in landslide susceptibility modelling. This modifies the fundamental paradigm of susceptibility in two aspects: first, spatial and temporal effect of landslide path dependency is an "add-on" dependent variable that has to be considered in combination with independent environmental factors. Second, landslide susceptibility is not time-invariant but instead is dynamic and changes over time. In our study area, this effect was obvious, and had a strong effect on the performance of landslide susceptibility models. However, this new paradigm is in its initial stage and has to be explored and studied in other landslide prone |

| | areas where detailed multi-temporal landslide inventories are available. We are aware that such multi-temporal inventories are rare but given the substantial progresses made already in remote sensing imagery and techniques, and in near future, this will provide room for creation of more multi-temporal landslide inventories where we believe would be main future direction in the field of landslide mapping and documenting. |
|---|---|
| Results and discussion

16. Figures 7, 8 and 9:

- the maps would be easier to interpret with a hillshade in the background and the outlines of the corresponding training and/or test landslides, which are required to assess the spatial performance of the models. | Thank you for this point. We have added the hillshade to the backgrounds of landslide susceptibility maps in the revised manuscript. However, it would not be helpful to show the outlines of landslide because there are too many landslides and they completely cover the area. |
| - it is a good idea to show the distribution of the susceptibility classes, but pie charts are not very effective for comparing multiple part-to-whole relationships. They are inconvenient to read and it is hard to perceive the quantitative relationships. Bar or column charts would be more suitable. | We appreciate the different preference but we feel that pie charts work better for this study. |
| - why are there blank/white areas in the maps containing path dependent variables? | The reason is that the model does not make a prediction for those areas because one of the explanatory factors contains no data in those areas. This factor is topographic position index (TPI) and it has no data in those areas because the slope is zero. The issue only occurs in the conventional plus path dependent model because only that model uses TPI as explanatory factor. It is easy to assign values of zero to TPI in those areas and then calculate the model with that manually changed TPI but we felt that would not be fair. By using TPI and the conventional plus path dependent model with no data we are underestimating model performance because no landslides occurred in any of no data areas and predictions would be zero if we assign zero values to TPI. |
| Results and discussion

17. L 345: I think this should be the map on the left in Figure 7. | Thank you for your sharp observation. This has been corrected in the updated manuscript. |
| Results and discussion

18. L 347-349, usage of landslide susceptibility maps for amount of time of landslide inventory: I think this is hard to generalize and depends on the task, but for sustainable planning of resilient urban areas I would rather counsel time-insensitive susceptibility models based on intrinsic parameters. | We certainly agree with you. The usage of landslide susceptibility maps indeed depends on the goal and task of audience. We also fully agree with you that for sustainable planning the conventional static landslide susceptibility maps are more useful. However, we believe that the dynamic path dependent landslide susceptibility maps would be also useful both in long-term and short-term planning. The dynamic path |

| | |
|---|---|
| | dependent landslide susceptibility maps consist of a static part taking intrinsic factors into account, and a dynamic part taking landslide path dependency into account. As you also have mentioned, the susceptibility maps in the static part are optimal for sustainable planning. However, if it's possible to compute the full dynamic susceptibility maps as in the Collozzone area, then this would be even more practical both for short-term and long-term planning. Another important point is that the only dynamic part of path dependent landslide susceptibility maps would be also in the interest of short-term land use planners and farmers where they would be aware of the areas with different intensity of landslide susceptibility.

As we discussed above, we have updated that part of manuscript in the revised version as following:

"The usage of conventional static landslide susceptibility maps and dynamic landslide susceptibility maps taking landslide path dependency depends on the goal and task of audience. In reality, static susceptibility maps created (either with a conventional susceptibility model, or as the static portion of a conventional plus path dependent model) can be used in sustainable planning whereas dynamic susceptibility maps can be considered in short-term land use planning." |
| 19. Figure 10: I do not understand what hypothetical means. Is this graph based on real data or is this just a sketch? | This is just a hypothetical sketch. We have updated this in the revised manuscript as following:

[revised manuscript text omitted]